# Clinical Outcome of Patients with Pelvic and Retroperitoneal Bone and Soft Tissue Sarcoma: A Retrospective Multicenter Study in Japan

**DOI:** 10.3390/cancers14123023

**Published:** 2022-06-20

**Authors:** Toshiyuki Takemori, Teruya Kawamoto, Hitomi Hara, Naomasa Fukase, Shuichi Fujiwara, Ikuo Fujita, Takuya Fujimoto, Masayuki Morishita, Kazumichi Kitayama, Shunsuke Yahiro, Tomohiro Miyamoto, Masanori Saito, Jun Sugaya, Katsuhiro Hayashi, Hiroyuki Kawashima, Tomoaki Torigoe, Tomoki Nakamura, Hiroya Kondo, Toru Wakamatsu, Munenori Watanuki, Munehisa Kito, Satoshi Tsukushi, Akihito Nagano, Hidetatsu Outani, Shunichi Toki, Shunji Nishimura, Hiroshi Kobayashi, Itsuo Watanabe, Yusuke Demizu, Ryohei Sasaki, Takumi Fukumoto, Takahiro Niikura, Ryosuke Kuroda, Toshihiro Akisue

**Affiliations:** 1Department of Orthoapedic Surgery, Graduate School of Medicine, Kobe University, Kobe 650-0017, Japan; westlife_take@yahoo.co.jp (T.T.); mitohi@med.kobe-u.ac.jp (H.H.); nfukase@msn.com (N.F.); fujiwarashuuichi@yahoo.co.jp (S.F.); kazoo_728@yahoo.co.jp (K.K.); syahiro59@yahoo.co.jp (S.Y.); bfm437@yahoo.co.jp (T.M.); tniikura@med.kobe-u.ac.jp (T.N.); kurodar@med.kobe-u.ac.jp (R.K.); akisue@med.kobe-u.ac.jp (T.A.); 2Department of Orthopaedic Surgery, Hyogo Cancer Center, Akashi 673-8558, Japan; fujita_19@hyogo-cc.jp (I.F.); fujitaku@hp.pref.hyogo.jp (T.F.); msmorishita_kch@hp.pref.hyogo.jp (M.M.); 3Division of Orthopaedic Surgery, Kobe University Hospital, International Clinical Cancer Research Center, Kobe 650-0047, Japan; 4Department of Orthopaedic Oncology, Cancer Institute Hospital, Japanese Foundation for Cancer Research, Tokyo 135-8550, Japan; masanori.saito@jfcr.or.jp; 5Department of Musculoskeletal Oncology, National Cancer Center Hospital, Tokyo 104-0045, Japan; jsugaya@east.ncc.go.jp; 6Department of Orthopaedic Surgery, Graduate School of Medical Sciences, Kanazawa University, Kanazawa 920-8641, Japan; hysk@med.kanazawa-u.ac.jp; 7Division of Orthopaedic Surgery, Department of Regenerative and Transplant Medicine, Graduate School of Medical and Dental Sciences, Niigata University, Niigata 951-8510, Japan; inskawa@med.niigata-u.ac.jp; 8Department of Orthopaedic Oncology and Surgery, Saitama Medical University International Medical Center, Hidaka 350-1298, Japan; ttorigoe@saitama-med.ac.jp; 9Department of Orthopaedic Surgery, Graduate School of Medicine, Mie University, Tsu 514-8507, Japan; tomoki66@clin.medic.mie-u.ac.jp; 10Department of Orthopaedic Surgery, Graduate School of Medicine, Dentistry, and Pharmaceutical Science, Okayama University, Okayama 700-8558, Japan; me20034@s.okayama-u.ac.jp; 11Department of Musculoskeletal Oncology Service, Osaka International Cancer Institute, Osaka 541-8567, Japan; evolutionhhh49@yahoo.co.jp; 12Department of Orthopaedic Surgery, School of Medicine, Tohoku University, Sendai 980-8574, Japan; mwata@ortho.med.tohoku.ac.jp; 13Department of Orthopaedic Surgery, School of Medicine, Shinshu University, Matsumoto 390-8621, Japan; mune0527@yahoo.co.jp; 14Department of Orthopaedic Surgery, Aichi Cancer Center Hospital, Nagoya 464-0021, Japan; s-tsuku@aichi-cc.jp; 15Department of Orthopaedic Surgery, Gifu University, Gifu 501-1194, Japan; a-nagano@lucky.odn.ne.jp; 16Department of Orthopaedic Surgery, Graduate School of Medicine, Osaka University, Suita 565-0871, Japan; h-otani@ort.med.osaka-u.ac.jp; 17Department of Orthopaedics, Institute of Biomedical Sciences, Graduate School, Tokushima University, Tokushima 770-8503, Japan; c000029043@tokushima-u.ac.jp; 18Department of Orthopaedic Surgery, Kindai University Hospital, Osaka-Sayama 589-8511, Japan; shunnisi@med.kindai.ac.jp; 19Department of Orthopaedic Surgery, Graduate School of Medicine, The University of Tokyo, Tokyo 113-8655, Japan; hkobayashi-tky@umin.ac.jp; 20Department of Orthopaedic Surgery, Tokyo Dental College Ichikawa General Hospital, Ichikawa 272-8513, Japan; itsuowatanabe@tdc.ac.jp; 21Department of Radiation Oncology, Hyogo Ion Beam Medical Center Kobe Proton Center, Kobe 650-0047, Japan; y_demizu@nifty.com; 22Division of Radiation Oncology, Graduate School of Medicine, Kobe University, Kobe 650-0017, Japan; rsasaki@med.kobe-u.ac.jp; 23Department of Surgery, Division of Hepato-Biliary-Pancreatic Surgery, Graduate School of Medicine, Kobe University, Kobe 650-0017, Japan; fukumoto@med.kobe-u.ac.jp; 24Department of Rehabilitation Science, Graduate School of Health Sciences, Kobe University, Kobe 654-0142, Japan

**Keywords:** pelvis, retroperitoneum, bone and soft tissue sarcoma, prognosis, prognostic factors

## Abstract

**Simple Summary:**

We aimed to clarify the clinical outcomes of patients with pelvic and retroperitoneal bone and soft tissue sarcoma. The 3-year overall survival (OS), local control (LC) rate, and progression-free survival (PFS) were 71.7%, 79.1%, and 48.6%, respectively. The most influential poor prognostic factor for OS was distant metastasis, and for PFS, this was higher age (≥60 years). Larger primary tumor size (≥10 cm) was the only poor prognostic factor for LC. In the histological analysis, osteosarcoma showed significantly worse OS and PFS than other sarcomas in the pelvis and retroperitoneum.

**Abstract:**

This study aimed to retrospectively analyze the clinical outcomes of patients with pelvic and retroperitoneal bone and soft tissue sarcoma (BSTS). Overall, 187 patients with BSTS in the pelvis and retroperitoneal region treated at 19 specialized sarcoma centers in Japan were included. The prognostic factors related to overall survival (OS), local control (LC), and progression-free survival (PFS) were evaluated. The 3-year OS and LC rates in the 187 patients were 71.7% and 79.1%, respectively. The 3-year PFS in 166 patients without any distant metastases at the time of primary tumor diagnosis was 48.6%. Osteosarcoma showed significantly worse OS and PFS than other sarcomas of the pelvis and retroperitoneum. In the univariate analyses, larger primary tumor size, soft tissue tumor, distant metastasis at the time of primary tumor diagnosis, P2 location, chemotherapy, and osteosarcoma were poor prognostic factors correlated with OS. Larger primary tumor size, higher age, soft tissue tumor, chemotherapy, and osteosarcoma were poor prognostic factors correlated with PFS in patients without any metastasis at the initial presentation. Larger primary tumor size was the only poor prognostic factor correlation with LC. This study has clarified the epidemiology and prognosis of patients with pelvic and retroperitoneal BSTS in Japan.

## 1. Introduction

Bone and soft tissue sarcomas (BSTSs) are uncommon and heterogeneous cancers of mesenchymal origin, representing approximately 1% of all cancers in the adult population [1]. BSTSs occur at various anatomical sites and have a wide variety of histological subtypes. The most common site of BSTSs is the extremities, and pelvic occurrences of malignant bone tumors have been reported to be only 16% [2]. Moreover, sarcomas in the retroperitoneal region have been reported to be rare tumors, accounting for approximately 15% of all soft tissue sarcomas [3]. The standard treatment for BSTSs is surgical resection. However, pelvic and retroperitoneal tumors are often difficult to resect surgically due to anatomical complexity [4]. Therefore, the prognosis of patients with pelvic and retroperitoneal BSTS has been reported to be poor compared to tumors in the extremities [5,6]. However, to the best of our knowledge, there are no reports on the prognosis and prognostic factors of patients with pelvic and retroperitoneal BSTS.

Therefore, we aimed to conduct a retrospective, multicenter study to evaluate the clinical outcomes of patients with pelvic and retroperitoneal BSTS in Japan.

## 2. Materials and Methods

### 2.1. Study Design and Patients

This was the first retrospective, multicenter study involving 19 specialized sarcoma centers in Japan. This study was approved by the Ethics Review Board of each institution. Informed consent was obtained in the form of an opt-out, and patients who rejected participation in the present study were excluded. We identified 218 patients with pelvic and retroperitoneal BSTS who were diagnosed in each institution between January 2012 and December 2016. Thirty-one patients with inadequate clinical records were excluded; the remaining 187 patients were included in the present study, and their medical records were retrospectively analyzed. Data extracted from each patient’s medical record included: age at primary tumor diagnosis, sex, primary tumor size in the greatest diameter, primary tumor location, primary tumor type (bone or soft tissue), histological subtype of the primary tumor, staging, follow-up period, detailed treatment information, presence of distant metastasis at the time of primary tumor diagnosis (M0 or M1), local recurrence, distant metastasis, and status at the last follow-up. The location of bone tumors was determined according to the Enneking classification [7]. The location of soft tissue tumors was also determined using a modified classification for soft tissue tumor based on the Enneking classification. Staging was classified according to the Enneking surgical staging system [8]. Overall survival (OS) and local control (LC) were defined as the duration between the date of the initial treatment and death, and regrowth or recurrence at or near primary tumor location, respectively. Progression-free survival (PFS) was defined as the duration between the date of the initial treatment and local recurrence or distant metastasis at any sites. With regard to PFS, we analyzed 166 patients without any distant metastases at the time of primary tumor diagnosis (M0). Clinical parameters with prognostic effects were analyzed, and univariate and multivariate analyses were used to identify factors associated with the clinical outcomes.

### 2.2. Statistical Analyses

OS, LC, and PFS curves were estimated using the Kaplan–Meier method, and the log-rank test was used to assess the differences in survival [9]. Differences and correlations were considered statistically significant at *p* < 0.05, and variables with a *p* < 0.05 from the univariate analysis were included in the multivariate analysis, using a Cox proportional hazards model. All statistical analyses were performed with EZR version 1.53 (Saitama Medical Center, Jichi Medical University, Saitama, Japan), a graphical user interface for R (The R Foundation for Statistical Computing, Vienna, Austria) [10].

## 3. Results

### 3.1. Patients

The present study included 114 males and 73 females, with a median age of 58 years (range 8–88 years) at the time of primary tumor diagnosis. The median primary tumor size in the greatest diameter was 10 cm (range 3–31 cm). The median follow-up period was 48 months (range of 1–106 months). The location of primary tumors based on the Enneking classification is shown in Table 1 [7]. Tumors located in region P2 that involved the acetabulum, a reportedly difficult-to-treat surgical site [11], were found in 52 patients (27.8%). Bone tumors accounted for 147 patients (78.6%) and soft tissue tumors for 40 patients (21.4%). One hundred and sixty-six patients (88.8%) were enrolled in M0, and 21 patients (11.2%) were enrolled in M1. The number patients in each stage, according to the Enneking surgical staging system, was 7 patients in ⅠA, 19 patients in ⅠB, 20 patients in ⅡA, 120 patients in ⅡB, and 21 patients in Ⅲ. Distant metastasis had occurred in 71 patients out of 161 M0 patients (42.8%) during the follow-up period. Ten out of one hundred and forty-seven patients (6.8%) with bone tumors and eleven out of forty patients (27.5%) with soft tissue tumors were enrolled in M1. Distant metastasis at the time of primary tumor diagnosis was significantly more frequent in patients with soft tissue tumors than in patients with bone tumors (*p* < 0.001).

The histological subtypes of the primary tumors are listed in Table 2. The most common subtype of primary tumor was chordoma, which presented in 54 patients (28.9%), followed by chondrosarcoma (34 patients, 18.2%), osteosarcoma (32 patients, 17.1%), liposarcoma (15 patients, 8.0%), undifferentiated pleomorphic sarcoma/malignant fibrous histiocytoma (14 patients, 7.5%), and Ewing sarcoma (11 patients, 5.9%).

Treatments for the primary tumor are listed in Table 3. Of the 95 patients who underwent surgical treatment for the primary lesion, 54 patients underwent surgery alone, 29 underwent surgery and chemotherapy, 6 underwent surgery and radiotherapy, and another 6 underwent surgery, chemotherapy, and radiotherapy. Of the 73 patients who underwent curative radiotherapy for the primary lesion, 49 underwent radiotherapy alone, and 24 underwent radiotherapy and chemotherapy. The curative radiotherapy was conducted using carbon ions (62 patients), protons (7 patients), or photons (4 patients). Treatments for the primary tumor in patients without curative surgery or radiotherapy were chemotherapy alone in seven patients, chemotherapy and palliative radiotherapy in eight patients, and palliative radiotherapy alone in three patients. One patient did not receive any treatments. In total, 74 patients (39.6%) were treated with chemotherapy, and 55 patients of the 166 M0 patients (33.1%) and 19 patients of 21 M1 patients (90.5%) received chemotherapy. By histological subtypes, chemotherapy was administered in 26 patients of the 32 osteosarcoma patients (81.3%) and 48 of the 155 patients with tumors other than osteosarcoma (31.0%). The administration of chemotherapy was significantly more frequent in M1 and osteosarcoma patients compared to those in M0 and patients with tumors other than osteosarcoma (*p* < 0.001).

### 3.2. Survival and Local Control

Of the 187 patients, the 3-year OS and LC rates were 71.7% (95% confidence interval (CI) 64.6–77.7%), and 79.1% (95% CI 71.9–84.7%), respectively (Figure 1a,b). Of the 166 M0 patients, the 3-year PFS was 48.6% (95% CI 40.8–56.0%) (Figure 1c). In addition, of the 166 M0 patients, local recurrence occurred in 37 patients (22.3%) at a mean of 19 months (range of 0–76 months) and distant metastasis occurred in 71 patients (42.8%) at a mean of 14.5 months (range of 0–78 months) after the initial treatment. In the histological analyses, the 3-year OS rates for chordoma, chondrosarcoma, and osteosarcoma were 93.9% (95% CI 82.3–98.0%), 85.2% (95% CI 68.0–93.6%), and 49.1% (95% CI 30.8–65.1%), respectively (Figure 2a). The 3-year LC rates for chordoma, chondrosarcoma, and osteosarcoma were 80.7% (95% CI 67.1–89.1%), 90.2% (95% CI 72.5–96.7%), and 70.1% (95% CI 46.2–84.9%), respectively (Figure 2b). Regarding LC rates, no significant difference among the three histological subtypes was observed. The 3-year PFS rates for chordoma, chondrosarcoma, and osteosarcoma were 56.1% (95% CI 41.7–68.3%), 73.9% (95% CI 54.5–86.0%), and 24.1% (95% CI 10.7–40.5%), respectively (Figure 2c). Osteosarcoma had significantly worse OS and PFS compared to chordoma (*p* < 0.001) and chondrosarcoma (*p* < 0.001), respectively.

The prognostic factors found via univariate and multivariate analyses are shown in Table 4 and Table 5. The logrank test revealed that a larger primary tumor size of 10 cm or more; soft tissue tumors; M1; P2 location; chemotherapy; osteosarcoma; and staging in ⅡA, ⅡB, and Ⅲ were significantly associated with a worse OS. Only a larger primary tumor size of 10 cm or more was detected as a significantly correlated factor with worse LC using univariate analysis. In addition, a larger primary tumor size (≥10 cm); higher age (≥60 years); soft tissue tumor; chemotherapy; osteosarcoma; and staging in ⅡA, ⅡB, and Ⅲ were significantly associated with a worse PFS in M0 patients. The Cox proportional hazards models revealed that a larger primary tumor size (≥10 cm), soft tissue tumor, M1, and osteosarcoma were significantly associated with worse OS; higher age (≥60 years), soft tissue tumor, and osteosarcoma were significantly associated with worse PFS in M0 patients. M1 was the most influential poor prognostic factor for OS among the 187 patients, while higher age (≥60 years) was the most influential poor prognostic factor for PFS among the 166 M0 patients.

## 4. Discussion

In the present study, we retrospectively analyzed the clinical outcomes and associated factors of the prognosis of patients with pelvic and retroperitoneal BSTSs. The multicenter profiles of the characteristics of 187 patients with pelvic and retroperitoneal BSTS were clarified. Furthermore, we provided an overview of the prognosis and significant factors affecting OS, LC, and PFS. To the best of our knowledge, this is the first multicenter study on BSTS arising in the pelvis and retroperitoneum in Japan that contains detailed data on epidemiology and treatment options.

BSTSs are very rare tumors, representing approximately 1% of all cancers in the adult population [1]. The most common site where BSTSs occur is the extremities, and BSTSs in the pelvis and retroperitoneum are less common. Approximately 16% of all malignant bone tumors and 15% of all soft tissue sarcomas have been reported to occur in the pelvis or retroperitoneum [2,3]. Thus, due to the small number of patients, there are no coherent reports on patients with pelvic and retroperitoneal BSTS.

The standard treatment for BSTS is surgical resection. Except of a few histological subtypes, most BSTSs are resistant to radiotherapy and/or chemotherapy. Since April 2016, carbon ion radiotherapy has been available for unresectable/inoperable BSTS in Japan. Following that, proton beam therapy has also been available since April 2018, and in recent years, clinically favorable results for particle beam therapy (carbon ion and proton beam) with regard to unresectable/inoperable BSTSs have been reported [12,13,14,15,16,17], especially for chordomas [12,18,19,20]. However, the long-term outcome of particle beam therapy for BSTS is unclear; therefore, the standard treatment for BSTS, except for chordoma, is still surgical resection. Surgery for BSTSs in the pelvis and retroperitoneum are often difficult due to anatomical complexity [4]. Even if complete tumor resection is possible, the reconstruction of skin or the hip joint may be required after tumor resection, resulting in a long operation time and high incidence of postoperative complications [4,21,22]. Therefore, patients with pelvic BSTS are reported to have a worse prognosis compared to the those with BSTS in the extremities [5]. In our study, the 3-year OS of patients with pelvic and retroperitoneal BSTS was 71.7%. In a report on 3826 patients with soft tissue sarcomas in Japan, the 2-year and 5-year disease-specific survival rates were reported to be 86.8% and 77.5%, respectively [6]. In comparison to the findings from our study, this suggests that the prognosis of patients with pelvic and retroperitoneal BSTS is poorer. Gronchi et al., in a review of 1007 retroperitoneal sarcoma patients, reported a 5-year OS of 67%, similar to our study [23]. In terms of histology, osteosarcoma had a significantly worse prognosis in the present study. Bielack et al., in a review of 1702 osteosarcoma patients, reported a 5-year OS of 28.9% for patients with pelvic osteosarcoma; this is significantly worse compared to that of patients with extremity osteosarcoma (67.3%) [24]. Other studies have reported 5-year survival rates for pelvic osteosarcoma of 18–34% [25,26,27]. Consistent with these reports, osteosarcoma was shown to have a poorer prognosis among the bone sarcomas.

In the present study, multivariate analyses revealed that a larger primary tumor size of 10 cm or more, soft tissue tumors, M1, and osteosarcoma were significantly associated with a worse OS. Moreover, higher age (≥60 years), soft tissue tumor, chemotherapy, and osteosarcoma were significantly associated with a worse PFS in M0 patients. In general, M1 [28,29], larger primary tumor size [30,31,32,33,34], higher age [33,34], and high-grade [28,29,30,31,32,33,34] have been reported as the poor prognostic factors of BSTS. Similarly, the present study identified these prognostic factors in pelvic and retroperitoneal BSTS. In addition, we revealed that soft tissue tumors and chemotherapy were poor prognostic factors. As far as we have examined, we found no article that has compared the prognosis between patients with bone and soft tissue sarcoma, or any comparison between these patients with and without chemotherapy. In the present study, chordoma and chondrosarcoma accounted for 88 patients of the 147 bone tumors (59.9%), and they had a better prognosis than the other histological subtypes. In contrast, most soft tissue sarcomas in the current study were of relatively high-grade histological subtypes. In addition, chemotherapy is usually administered to patients with high-grade tumors who have poor prognoses, except for patients with a low-grade and/or slowly-growing tumors, such as chordoma and chondrosarcoma, which are generally resistant to chemotherapy [35,36]. These factors may influence the current findings with respect to soft-tissue tumors and chemotherapy that were explored as independent factors associated with OS and/or PFS in univariate or multivariate analyses.

The present study has several limitations. First, due to its retrospective design, we cannot exclude the possibility of selection bias. Second, the follow-up period was relatively short (median of 48 months). Therefore, we hope to continue the observation of these patients and to report on their follow-ups in future studies. Finally, particle beam therapy has represented a definitive treatment for unresectable/inoperable BSTS in Japan recently. Further studies are needed to clarify the prognosis and prognostic factors depending on treatment methods.

## 5. Conclusions

The current study clarified that the prognosis of patients with pelvic and retroperitoneal BSTS is poor. This study may be useful in predicting appropriate prognoses and planning treatments for patients with pelvic and retroperitoneal BSTS.

## Figures and Tables

**Figure 1 cancers-14-03023-f001:**
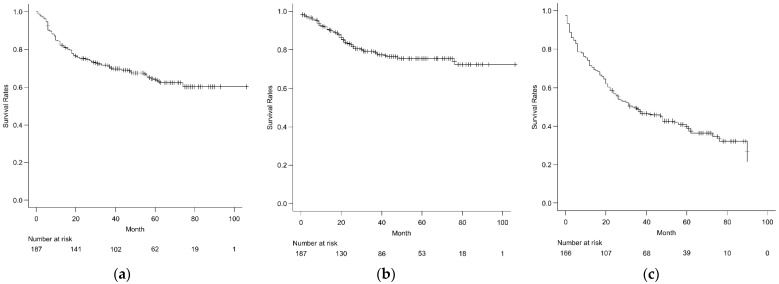
(**a**) Overall survival in 187 patients with pelvic and retroperitoneal bone and soft tissue sarcoma; (**b**) local-control rates in 187 patients with pelvic and retroperitoneal bone and soft tissue sarcoma; (**c**) progression-free survival in 166 M0 patients with pelvic and retroperitoneal bone and soft tissue sarcoma.

**Figure 2 cancers-14-03023-f002:**
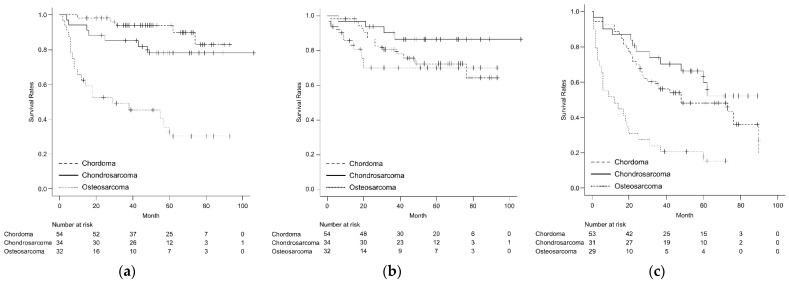
(**a**) The comparison of overall survival among the patients with chordoma, chondrosarcoma, and osteosarcoma; (**b**) the comparison of local control rates among the patients with chordoma, chondrosarcoma, and osteosarcoma; (**c**) the comparison of progression-free survival among the patients with chordoma, chondrosarcoma, and osteosarcoma.

**Table 1 cancers-14-03023-t001:** Primary tumor locations.

Locations	*N* (%)
P1	47 (25.1)
P1, 2	23 (12.3)
P1, 4	11 (5.9)
P1, 2, 3	4 (2.1)
P1, 2, 4	4 (2.1)
P1, 3, 4	1 (0.5)
P1, 2, 3, 4	3 (1.6)
P2	3 (1.6)
P2, 3	14 (7.5)
P2, 4	1 (0.5)
P3	14 (7.5)
P4	62 (33.2)
Total	187 (100)

P1: ilium; P2: acetabulum; P3: ischium; P4: sacrum.

**Table 2 cancers-14-03023-t002:** Histological subtypes of primary tumors.

Tissue	Histological Subtypes	*N* (%)
Bone	Chordoma	54 (28.9)
	Chondrosarcoma	34 (18.2)
	Osteosarcoma	32 (17.1)
	Ewing sarcoma	11 (5.9)
	UPS/MFH	7 (3.7)
	Others	9 (4.8)
Soft tissue	Liposarcoma	15 (8.0)
	UPS/MFH	7 (3.7)
	MPNST	6 (3.2)
	Leiomyosarcoma	4 (2.1)
	Others	8 (4.3)
	Total	187 (100)

UPS/MFH: undifferentiated pleomorphic sarcoma/malignant fibrous histiocytoma; MPNST: malignant peripheral nerve sheath tumor.

**Table 3 cancers-14-03023-t003:** (**a**) Treatments for primary tumor. (**b**) Treatment details of surgery and curative radiotherapy.

**(a)**		
Treatment		*N* (%)
Surgery		95 (50.8)
	Surgery alone	54
	Surgery + Chemotherapy	29
	Surgery + Palliative Radiotherapy	6
	Surgery + Chemotherapy + Palliative Radiotherapy	6
Curative Radiotherapy		73 (39.0)
	Curative Radiotherapy alone	49
	Curative Radiotherapy + Chemotherapy	24
Chemotherapy		15 (8.0)
	Chemotherapy alone	7
	Chemotherapy + Palliative Radiotherapy	8
Palliative Radiotherapy		3 (1.6)
Best Supportive Care		1 (0.5)
Total		187 (100)
**(b)**		
Treatment		*N*
Surgery		95
	Wide resection	70
	Marginal resection	13
	Intra-tumoral resection	4
	Amputation	7
	Unknown	1
Curative Radiotherapy		73
	Carbon ion	62
	Protons	7
	Photons	4

**Table 4 cancers-14-03023-t004:** Univariate and multivariate analyses for OS and LC.

	OS				LC	
		Uni	Multi			Uni
Characteristics (*N*)	3y (%)	*p*	*p*	HR	3y (%)	*p*
				(95% CI)		
Age (years)						
≥60 (86)	68.1	0.182			81.3	0.679
<60 (101)	74.8				77.7	
Sex						
Male (114)	68.7	0.267			76.9	0.253
Female (73)	76.4				82.5	
Tumor size (cm)						
≥10 (96)	58.5	**<0.001**	**0.021**	0.515	70.5	**0.009**
<10 (91)	85.5			(0.293–0.907)	86.8	
Tumor location						
With P2 (52)	57.0	**0.009**	0.091		80.8	0.752
Without P2 (135)	77.4				78.9	
Tumor type						
Bone (147)	76.5	**0.008**	**0.048**	1.943	81.2	0.253
Soft tissue (40)	53.9			(1.005–3.756)	71.4	
Metastasis						
M0 (166)	78.0	**<0.001**	**<0.001**	5.365	78.5	0.412
M1 (21)	17.9			(2.556–11.26)	91.7	
Chemotherapy						
Yes (74)	56.4	**<0.001**	0.298		76.1	0.389
No (113)	81.8				81.1	
Histological subtypes						
Osteosarcoma (32)	49.1	**<0.001**	**<0.001**	4.000	70.1	0.273
Others (155)	76.4			(1.955–8.182)	80.7	
Staging						
IA, IB (26)	91.5				80.4	0.806
IIA, IIB, III (161)	68.5	**0.004**	0.089		79.2	

OS: overall survival; LC: local control; HR: hazard ratio; CI: confidence interval. Bold number indicates *p* value smaller than 0.05.

**Table 5 cancers-14-03023-t005:** Univariate and multivariate analyses for PFS in M0 patients.

	PFS			
		Uni	Multi	
Characteristics (*N*)	3y (%)	*p*	*p*	HR
				(95% CI)
Age (years)				
≥60 (79)	41.5	**0.046**	**0.002**	0.527
<60 (87)	55.0			(0.351–0.790)
Sex				
Male (99)	43.4	0.243		
Female (67)	56.4			
Tumor size (cm)				
≥10 (78)	35.9	**0.003**	0.051	
<10 (88)	59.8			
Tumor location				
With P2 (43)	44.1	0.091		
Without P2 (123)	50.3			
Tumor type				
Bone (137)	52.3	**0.018**	**0.004**	2.118
Soft tissue (29)	31.0			(1.277–3.513)
Chemotherapy				
Yes (55)	30.9	**<0.001**	0.124	
No (111)	57.4			
Histological subtypes				
Osteosarcoma (29)	24.1	**<0.001**	**0.006**	2.170
Others (137)	53.8			(1.246–3.780)
Staging				
IA, IB (26)	69.2			
IIA, IIB (140)	44.8	**0.014**	0.064	

PFS: progression-free survival, HR: hazard ratio, CI: confidence interval. Bold number indicates *p* value smaller than 0.05.

## Data Availability

No new data were created or analyzed in this study. Data sharing is not applicable to this article.

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
