# Peer review of "Clinical Outcome of Patients with Pelvic and Retroperitoneal Bone and Soft Tissue Sarcoma: A Retrospective Multicenter Study in Japan"

_cancers, 2022, doi:10.3390/cancers14123023_

Round 1
Reviewer 1 Report
Sir,
I read the paper by Professor Toshiyuki Takemori and coll.
I recognise the great effort to collect all those data from different institutions in Japan, but some shortcomings ere evident:
1) This is a retrospective collection of data from 2012 to 2016. Some aspects of the surgical approach (i.e multivisceral resection in retroperitoneal sarcomas RPS) have changed during this time
2) RPS and bone sarcoma of the pelvis , in Europe at least , are treated in referral centers considering the difficulty in the treatment and the lack of expertise on those rare neoplasms. Probably in Japan the organization is different, but are we sure that all the centers were at the same level of expertise? Were the results biased by the multicenter approach with different level of expertise?
3) Looking at the mix of cases as expert in sarcomas I have some doubts:
a) RPS in 65% of the cases are situated at least in origin at renal level, very far from pelvis
b) RPS rarely involves the bone structures,. So the surgical approach are different in RPS ( see Bonvalot JCO 2009) and in bone sarcomas of the pelvis ( see Enneking as correctely reported in the text). In Europe and in USA RPS and bone sarcomas are usually treated by different staffs or in multidisciplinary teams including specialist of RPS and Orthopedics.
c) Histology are completely different: and can explain the different prognosis
RPS: 75% are Liposarcomas ( well differentiated, dedifferentiated, mixoid, undifferentiated) , 15% Leiomyosarcomas, then SFT, MPNST , rhabdo in children
Bone sarcomas: osteosarcomas ( with 10 different subtypes) Chondrosarcomas (G1,G2,G3) Chordomas, Ewing sarcomas.
All this sarcomas have a different natural history, different approaches, different sensitivity to Radio and chemotherapy:
Liposarcoma can be treated with preoperative RT (see STRASS study Bonvalot Lancet Oncology 2021), leiomyosarcoma can be treated with preoperative chemotherapy Ewing must be treated with preoperatory Chemo and radiotherapy ( see ESMO guidelines Annals of Oncology 2019) classical osteosarcoma recognizes an important role in chemotherapy but is totally resistant to radiotherapy. At last Chondrosarcome and Chordoma not operable can be treated with exclusive palliative radiotherapy but are absolutely resistant to chemotherapy.
We cannot consider in a unique report of cases all these different tumors.
4) In the discussion the Authors write that “To the best of our knowledge, this is the first multicenter study for BSTS arising in the pelvis and retroperitoneum in Japan that contains detailed data of epidemiology and treatment options”
This statement could be right in Japan, but many papers are published on treatment and prognosis of RPS ( Gronchi A. et al Ann Surg 2016 263 1002 -1009) Transatlantic cooperative group guidelines ( Ann Oncol 29 857-871 2019)
And on OS of the pelvis ( Brown DOI 2021 8027314 Sarcoma 2021) Donati EJSO 30 April 2004 322-340)
For the above mentioned reasons ( see point 3) nobody has ever considered RPS and bone sarcomas of the pelvis grouped altogether. About Chondrosarcoma see Pring JB &JS 2001;
The positive aspect of the paper is the epidemiological and comparative statistical analysis of the two groups of tumors where appears the different prognosis of the single histological
subtypes
Author Response
Dear Reviewer,
We wish to express our appreciation for your insightful comments and suggestions for our manuscript titled “Clinical Outcome of Patients with Pelvic and Retroperitoneal Bone and Soft Tissue Sarcoma: A Retrospective Multicenter Study in Japan”.
We greatly appreciated your time and efforts spent on the critical reading of our manuscript and providing feedback, which helped us greatly to improve our manuscript for publication. We have carefully revised the manuscript according to the Reviewers’ comments and prepared the point-by-point responses to the questions as outlined below. All the changes in the manuscript are highlighted in yellow.
Thank you very much for considering our paper for publication in the Cancers and we look forward to hearing from you.
Sincerely yours,
Teruya Kawamoto
Comment 1&2: This is a retrospective collection of data from 2012 to 2016. Some aspects of the surgical approach (i.e multivisceral resection in retroperitoneal sarcomas RPS) have changed during this time. RPS and bone sarcoma of the pelvis, in Europe at least, are treated in referral centers considering the difficulty in the treatment and the lack of expertise on those rare neoplasms. Probably in Japan the organization is different, but are we sure that all the centers were at the same level of expertise? Were the results biased by the multicenter approach with different level of expertise?
Response: Thank you very much for your invaluable comments. All patients included in this study were treated at specific facilities for sarcoma with specialized orthopaedic oncologists. And, all institutions regularly have a multidisciplinary discussion for sarcoma cases, and could collaborate with general surgeons, urologists, and other surgeons in cases of RPS and pelvic sarcomas. Therefore, we believe that there is little difference in level of expertise among the institutions.
Comment 3: a) RPS in 65% of the cases are situated at least in origin at renal level, very far from pelvis b) RPS rarely involves the bone structures. So the surgical approach are different in RPS (see Bonvalot JCO 2009) and in bone sarcomas of the pelvis (see Enneking as correctely reported in the text). In Europe and in USA RPS and bone sarcomas are usually treated by different staffs or in multidisciplinary teams including specialist of RPS and Orthopedics. c) Histology are completely different: and can explain the different prognosis RPS: 75% are Liposarcomas (well differentiated, dedifferentiated, mixoid, undifferentiated), 15% Leiomyosarcomas, then SFT, MPNST, rhabdo in children. Bone sarcomas: osteosarcomas (with 10 different subtypes) Chondrosarcomas (G1, G2, G3) Chordomas, Ewing sarcomas. All this sarcomas have a different natural history, different approaches, different sensitivity to Radio and chemotherapy: Liposarcoma can be treated with preoperative RT (see STRASS study Bonvalot Lancet Oncology 2021), leiomyosarcoma can be treated with preoperative chemotherapy Ewing must be treated with preoperatory Chemo and radiotherapy ( see ESMO guidelines Annals of Oncology 2019) classical osteosarcoma recognizes an important role in chemotherapy but is totally resistant to radiotherapy. At last Chondrosarcome and Chordoma not operable can be treated with exclusive palliative radiotherapy but are absolutely resistant to chemotherapy. We cannot consider in a unique report of cases all these different tumors.
Response: Thank you for your invaluable comments with detailed information. As you indicated, there are various types of sarcomas in this study, and due to the rarity and variety of subtypes of RPS and pelvic sarcomas, we agree that cases in this study should be different in histological subtypes, natural history and treatment approach. With regard to treatments, indications of pre/post-operatory chemotherapy and/or radiotherapy were determined by a multidisciplinary discussion for each patient at each institution, and we believe that appropriate treatment approach for each patient was performed. Moreover, in Japan, we can consider a particle radiation (carbon ion or proton) for inoperable pelvic and retroperitoneal BSTS, and the therapy should be effective in various sarcoma types.
Comment 4: In the discussion the Authors write that “To the best of our knowledge, this is the first multicenter study for BSTS arising in the pelvis and retroperitoneum in Japan that contains detailed data of epidemiology and treatment options”. This statement could be right in Japan, but many papers are published on treatment and prognosis of RPS (Gronchi A. et al Ann Surg 2016 263 1002 -1009) Transatlantic cooperative group guidelines (Ann Oncol 29 857-871 2019). And on OS of the pelvis (Brown DOI 2021 8027314 Sarcoma 2021) (Donati EJSO 30 April 2004 322-340). For the above mentioned reasons (see point 3) nobody has ever considered RPS and bone sarcomas of the pelvis grouped altogether. About Chondrosarcoma see Pring JB &JS 2001. The positive aspect of the paper is the epidemiological and comparative statistical analysis of the two groups of tumors where appears the different prognosis of the single histological subtypes.
Response: Thank you for your helpful comments. As your suggestions, there are several reports regarding as prognosis of RPS and as subtypes of bone and soft tissue sarcomas in pelvis. However, this study is the real-world treatment approach and clinical outcome of pelvic and retroperitoneal BSTS in Japan, and we believe that this should be very meaningful to clarify their treatment options and prognosis. In Discussion, we compared our results to previous reports (Ref# 23, 27 and Page8 Line255).
Reviewer 2 Report
Congratulations to authors for the big effort in conducting a multicenter trial in Japan to collect data on bone and soft tissue sarcomas, a rare disease.
The quality of data is good, the presentation is good.
The problem is that putting together bone pelvic sarcoma and retroperitoneal sarcoma, for outcome analysis, generates confusing data. Some considerations:
- the two groups are different in numbers (147 bone tumors, 40 retroperitoneal sarcomas)
- bone tumors and retroperitoneal sarcoma are different diseases, so I don't agree with the conclusion that data from this study are useful to predict prognosis of patients with sarcoma. In front of a patient you need to understand better the right prognosis of his disease, in order to plan the right treatment. It's not useful to know the prgnosis of a group of different entities.
In conclusion: I think that with your data it's better to make two separate analyses, one for the group of pelvic bone sarcoma, and one for the group of retroperitoneal sarcoma. Maybe you can write two different papers.
Some observations you could take into account when re-writing
- multicenter studies have pros and contra. In the case of rare tumors referral center are important. You have data of 218 pz in 19 center in 4 years. It could be good (to add value to your data) to say how many patients came from referral centers, and, if you think that there are selection biases, you could just describe them, or the reason why these occurred.
- It could be better to know tumor grade (or at least the difference between low-grade and high-grade tumors). Grade and stage are the 2 main prognostic factors in sarcomas
- in the group of bone tumors chondrosarcomas have better outcomes compared to osteosarcoma. Which histologic type of chondrosarcoma have you included? (dedifferentiated, low-grade or high-grade?)
- TABLE 1- better clarify in the legend what is P1-4.
- In the discussion, I would add a comparison of your outcome data with literature data
Author Response
Dear Reviewer,
We wish to express our appreciation for your insightful comments and suggestions for our manuscript titled “Clinical Outcome of Patients with Pelvic and Retroperitoneal Bone and Soft Tissue Sarcoma: A Retrospective Multicenter Study in Japan”.
We greatly appreciated your time and efforts spent on the critical reading of our manuscript and providing feedback, which helped us greatly to improve our manuscript for publication. We have carefully revised the manuscript according to the Reviewers’ comments and prepared the point-by-point responses to the questions as outlined below. All the changes in the manuscript are highlighted in yellow.
Thank you very much for considering our paper for publication in the Cancers and we look forward to hearing from you.
Sincerely yours,
Teruya Kawamoto
Comment 1: the two groups are different in numbers (147 bone tumors, 40 retroperitoneal sarcomas). Bone tumors and retroperitoneal sarcoma are different diseases, so I don’t agree with the conclusion that data from this study are useful to predict prognosis of patients with sarcoma. In front of a patient you need to understand better the right prognosis of his disease, in order to plan the right treatment. It’s not useful to know the prognosis of a group of different entities. In conclusion: I think that with your data it's better to make two separate analyses, one for the group of pelvic bone sarcoma, and one for the group of retroperitoneal sarcoma. Maybe you can write two different papers.
Response: Thank you for your invaluable comments. As you indicated, there are various types of sarcomas in this study, and due to the rarity and variety of subtypes of pelvic and retroperitoneal BSTS, cases in this study should be different in histological subtypes and treatment approach. However, this study is the real-world treatment approach and clinical outcome of pelvic and retroperitoneal BSTS in Japan, and we believe that this should be very meaningful to clarify their treatment options and prognosis.
Comment 2: Multicenter studies have pros and contra. In the case of rare tumors referral center are important. You have data of 218 pz in 19 center in 4 years. It could be good (to add value to your data) to say how many patients came from referral centers, and, if you think that there are selection biases, you could just describe them, or the reason why these occurred.
Response: Thank you for your insightful comment. Pelvic and retroperitoneal BSTS are extremely rare disease, therefore we believe that multicenter study is indispensable for analyzing such rare diseases. All patients included in this study were treated at specific facilities for sarcoma with specialized orthopaedic oncologists, and all institutions regularly have a multidisciplinary discussion for each case. Therefore, we believe that there is little difference in level of expertise among the institutions.
Comment 3: It could be better to know tumor grade (or at least the difference between low-grade and high-grade tumors). Grade and stage are the 2 main prognostic factors in sarcomas
Response: Thank you very much for your important recommendation. We employed the staging classified by Enneking surgical staging system as a factor (Page3 Line117). The number patients in each stage according to Enneking surgical staging system added in Page 3 Line 144. And, we re-analyzed the prognosis including the staging (Page5 Line199, 203, and Table 4, 5).
Comment 4: in the group of bone tumors chondrosarcomas have better outcomes compared to osteosarcoma. Which histologic type of chondrosarcoma have you included? (dedifferentiated, low-grade or high-grade?)
Response: Thank you for the comment. We re-analyzed with adding the staging according to Enneking surgical staging system as a factor (Please see the response to Comment 3). With regard to chondrosarcoma, we cannot provide the detailed histologic subtypes of chondrosarcoma, but the number of patients in each stage of 34 patients was one patient in IA, five in IB, five in IIA, 18 in IIB, and five in III.
Comment 5: TABLE 1- better clarify in the legend what is P1-4.
Response: We appreciate the suggestions. We have added the details of P1-4 classification in the legend of Table 1 (Page4).
Comment 6: In the discussion, I would add a comparison of your outcome data with literature data
Response: Thank you very much for your recommendation. Thank you for your helpful comments. As your suggestions, there are several reports regarding as prognosis of pelvic and retroperitoneal BSTS. In Discussion, we compared our results to previous reports (Ref# 23, 27 and Page8 Line255).
Round 2
Reviewer 1 Report
Some suggestions have been accepted.
Thank you.
The difficult comparison between RPS and bone sarcomas of the pelvis is still present.
The bibliography has been improved.
Author Response
Thank you very much for your invaluable comment. All retroperitoneal sarcomas included in this study arose from and mainly located in the pelvic cavity. Cases in which tumors located near kidneys in upper retroperitoneum were excluded, and we applied the Enneking classification for all tumor locations. The main objective in this study was to elucidate the clinical outcome of the patients with pelvic bone and soft tissue sarcomas in Japan. Therefore, we believe that retroperitoneal tumors located in the pelvic cavity should be lumped together.
Reviewer 2 Report
Thank you for editing your manuscript.
It's still not clear to me which were the primary tumor locations of the 40 soft tissue retroperitoneal and/or pelvic soft tissue sarcomas you included in your study. In my opinion, it is relevant information.
Author Response
Thank you very much for your important comment. All retroperitoneal sarcomas included in this study arose from and located in the pelvic cavity. Cases in which tumors mainly located near kidneys in upper retroperitoneum were excluded.